# Comparison of the Nasal Cavity Guidance Methods’ Effects during Nasotracheal Intubation Using a Preformed Nasotracheal Tube: A Prospective Randomized Controlled Trial

**DOI:** 10.3390/ijerph20054503

**Published:** 2023-03-03

**Authors:** Joungmin Kim, Eun-A Jang, Dongho Kang, Seonho Moon, Hong-Beom Bae

**Affiliations:** 1Department of Anesthesiology and Pain Medicine, Chonnam National University Hospital, Gwangju 61469, Republic of Korea; 2Department of Anesthesiology and Pain Medicine, Chonnam National University Medical School, Hwasun 58128, Republic of Korea; 3Department of Anesthesiology and Pain Medicine, Chonnam National University Hwasun Hospital, Hwasun 58128, Republic of Korea; 4Department of Anesthesiology and Pain Medicine, School of Dentistry, Chonnam National University, Gwangju 61186, Republic of Korea

**Keywords:** nasotracheal intubation, nasogastric tube, suction catheter, intubation time, airway complications

## Abstract

Nasotracheal intubation is mainly performed to provide a safe airway during maxillofacial surgeries. Several guiding devices are suggested to facilitate nasotracheal intubation and reduce complications. We attempted to compare intubation conditions during nasotracheal intubation using a nasogastric tube and a suction catheter, which are readily available in operating rooms. In this study, 114 patients undergoing maxillofacial surgery were randomly divided into the nasogastric tube guidance group (NG group) and the suction catheter guidance group (SC group). The primary outcome was the total intubation time. Moreover, the incidence and degree of epistaxis, the position of the tube in the nasal cavity after intubation, and the number of manipulations during intubation in the nasal cavity were investigated. The insertion time from the nostril to the oral cavity and the total intubation time were significantly shorter in the SC group than in the NG group (*p* < 0.001). The incidence of epistaxis was lower at 35.1% in the NG group and 43.9% in the SC group than the previously reported 60–80%, but there was no statistical difference between the two groups. The use of a suction catheter aid during nasotracheal intubation can be used effectively because it shortens the intubation time and does not increase complications.

## 1. Introduction

Nasotracheal intubation is performed to maintain the airway while securing an operating field during oral and maxillofacial surgeries [1]. However, during nasotracheal intubation, it is difficult to visually check the structures of the nasal cavity through which the tube passes. In the case of an anatomical change or a narrowing of the nasal cavity, difficult passage of the tube or trauma may result. During blind conventional intubation, complications such as epistaxis, obstruction of the nasotracheal tube, bacteremia, perforation of pyriform fossa, and sinusitis, etc., may occur as the intubation tube passes through the nasopharynx [2]. For the prevention and minimization of complications of nasotracheal intubation, the efficiency, speed, and stability of nasotracheal intubation are essential.

Several methods have been proposed to facilitate tube passage and minimize the complications of nasotracheal intubation [2,3,4,5,6,7,8,9,10]. Prior to intubation, it is necessary to carefully evaluate the anatomical variations and conditions of the nasal and oral cavities, and select a suitably sized tube for the patient. The use of an appropriate guiding device may further facilitate intubation [2]. A guiding device that easily passes the nasal cavity has the advantage of enabling the smooth passage of the nasotracheal tube while reducing the potential intranasal damage. Each of these guiding devices has different physical and morphological characteristics.

Among them, nasogastric tubes and suction catheters are easily available in operating rooms. Both have a long thin shape and good flexibility, which allows them to easily pass from the nasal cavity into the oropharynx.

We hypothesized that differences in nasotracheal intubation conditions may occur due to the different physical and morphological characteristics of these guiding devices. Therefore, this study aimed to investigate and compare the intubation conditions facilitated by these two guiding devices for nasotracheal intubation.

## 2. Materials and Methods

The study protocol was approved by the Institutional Review Board of Chonnam National University Hospital (CNUH-2020-242), and registered in the Clinical Research Information Service of the Korea National Institute of Health in the Republic of Korea (KCT0005987), which belongs to the World Health Organization Registry Network. Written informed consent was obtained from all the patients. Adult patients who were scheduled to undergo oral and maxillofacial surgery requiring nasotracheal intubation were included in the study. Patients with an American Society of Anesthesiologists (ASA) physical status >III, age <20 years, body mass index (BMI) ≥ 35, abnormal coagulopathy, oropharyngeal tumor, a known or predicted difficult airway, a history of nasal trauma, and previous basal skull fracture or surgery were excluded. The enrolled patients were randomly allocated to either the NG (nasogastric) group or SC (suction catheter) group, using a computer-generated randomization code; the allocation rate was 1:1. Pre-operatively, an airway examination was performed, including determining Mallampati grade (I–IV), mouth opening limitation, neck extension limitation, nasal or oral anatomical abnormalities, neck circumference, and thyromental distance. The nostrils were selected by examining the patient’s obstructive symptom and nasal airflow when breathing through the right and left nostrils alternately. Before the patient entered the operating room, the intubating equipment for each group was prepared which included Polar™ Preformed Tracheal Tubes (Smith Medical, Dublin, OH, USA), a Nasogastric Tube (12Fr, Yushin medical, Bucheon, Republic of Korea), or the Mallinckrodt Broncho-Cath Kit Suction Catheter (10Fr, Mallinckrodt, Raleigh, NC, USA). The preformed nasotracheal tube was initially selected based on gender (internal diameter 6.5 mm for females and 7.0 mm for males). When the patient arrived at the operating room, by comparing the diameter of the patient’s nostrils and the tube, a 0.5 mm smaller tube was prepared if the diameter of the nostril was small.

The patient was not administered pre-medication before entering the operating room. In the operating room, standard intraoperative monitoring equipment, such as electrocardiogram, pulse oximetry, non-invasive blood pressure measurement, and capnography were applied to all patients. All patients took the sniffing posture in the supine position. The height of the bed was adjusted so that the patient’s head was positioned between the intubator’s umbilicus and the epigastrium. Patients were preoxygenated with 100% oxygen at 3 min. General anesthesia was induced with propofol at 1.5~2.0 mg/kg, and 0.1~1.0 mcg/kg/min of remifentanil was injected according to the patient’s hemodynamic response. When the patient’s loss of consciousness was confirmed, 0.6 mg/kg of rocuronium was injected to facilitate intubation. In each group, a nasogastric tube or a suction catheter coated with water-soluble jelly was placed in a preformed nasotracheal tube lubricated with water-soluble jelly. If the guiding device is too long, it is difficult to remove it, and if it is too short, there is a possibility that the tube cannot be guided. As shown in Figure 1, the preformed nasotracheal tube was threaded over the prepared nasogastric tube (Figure 1A) or the suction catheter (Figure 1B) and protruded by 10 cm from the distal end of the preformed nasotracheal tube using the Seldinger technique.

The nasogastric tube or suction catheter together with the preformed nasotracheal tube was inserted into the nostril, and entered along the nasal floor toward the posterior nasopharyngeal wall. Once the suction catheter or the nasogastric tube passed through the nasal cavity to the vocal cord, the preformed nasotracheal tube was introduced into the nasal cavity through the guidance tool, and the guiding suction catheter or nasogastric tube was removed. The preformed nasotracheal tube was then intubated to the trachea through a direct laryngoscope. If necessary, the tube was introduced into the glottic inlet using Magill forceps. Proper intubation was confirmed by inflating the nasotracheal tube cuff, auscultation, and continuous monitoring of the end-tidal CO_2_.

The number of manipulations during intubation in the nasal cavity was measured using the following method. During intubation, a preformed nasotracheal tube was first inserted into the lower pathway (1st manipulation). If the nasotracheal tube could not enter due to resistance, the tube was redirected more caudally (2nd manipulation). If the tube still could not enter, it was slightly redirected in the cephalic direction (3rd manipulation). When the first intubation failed, intubation was attempted through the other nostril in the same way. Each insertion attempt was recorded. Moreover, the navigability (smooth, resistance, obstructed) required to insert the tube into the oropharynx from the nasal cavity was recorded. After a preformed nasotracheal tube entered the oropharynx, the degree of nasal bleeding was evaluated through a direct laryngoscope using a 4-point scale (no epistaxis; mild epistaxis: blood on the tracheal tube only; moderate epistaxis: blood pooling in the pharynx; or severe epistaxis: blood in the pharynx sufficient to impede intubation. The use of the Magill forceps, neck extension, tube rotation, backward, upward, and rightward pressure (BURP) manipulation, and Cormack grading of the glottis during direct laryngoscope insertion of the tube into the trachea were documented.

The insertion time from the nostril to the oropharynx and total intubation time were calculated by recording the time when the tip of the nasogastric tube or suction catheter began to be inserted into the nostril, the time when the tip of the nasotracheal tube entered the oropharynx, and the time when the first end-tidal CO_2_ was displayed on the monitoring screen. After tracheal intubation was completed, the tube location in the upper or lower pathway of the nasal cavity was checked using a fiberoptic endoscope. After the operation, the total anesthesia time and the operation procedures were recorded. Nasal pain was evaluated via a numerical rating scale (NRS, from 0 (no pain) to 10 (worst pain)) and complications such as persistent epistaxis were checked in the post-anesthesia care unit (PACU). We considered that persistent epistaxis was if nasal bleeding continued in the PACU after extubation of the nasotracheal tube. Patients usually stayed in the PACU for about an hour and were moved to the ward if they were included in the conditions for Post-Anesthetic Recovery Score.

The primary outcome of this study was the total intubation time. The secondary outcomes were the degree of nasal bleeding at intubation through a direct laryngoscope, the number of manipulations in the nasal cavity, navigability, the use of Magill forceps, nasotracheal tube pathway in the nasal cavity, nasal pain, and persistent epistaxis at PACU.

The possibility that physical characteristics such as bending deformation of the suction catheter and nasogastric tube may affect tube guidance was also considered. Accordingly, an investigation of this variable at Chonnam National University Engineering Practice Education Center was requested. A stress relaxation test was performed as the 3-point bending test using the DMA 2980 Dynamic Mechanical Analyzer. The relaxation modulus was measured for 10 min while applying 0.1% strain (change in shape and volume that occurs when an object is subjected to an external force) at 35 ℃ to the suction catheter and the nasogastric tube, respectively. The force applied while removing the aids from the preformed nasotracheal tube was also assessed. The force was measured 5 times for each preformed nasotracheal tube-guiding device combination. Three different sizes (6.00 mm, 6.5 mm, and 7.00 mm) of the nasotracheal tube were tested.

The sample size was calculated by measuring the total time required for tracheal intubation in a pilot study. The total time for tracheal intubation under suction catheter guidance was 53.73 ± 16.43 s, and under nasogastric tube guidance was 62.6 ± 15.09 s. Using G*Power 3.1.9.4, the sample size of 51 patients per group was calculated for 80% power and α-error of 5% (two-tailed test). Considering a dropout rate of 10%, it was calculated that 57 patients per group, and a total of 114 patients, were needed. SPSS for Windows ver. 21.0 (SPSS Inc., Chicago, IL, USA) was used for statistical analysis. The Shapiro–Wilk test was performed to examine the assumption of normality. Student’s *t*-test was used to compare normally distributed continuous variables, and the Mann–Whitney U test was used to compare non-normally distributed continuous variables and ordinal variables. Categorical variables were compared using the chi-square test. All the measured values are presented as the number of patients, mean ± standard deviation (SD), or median [25–75% interquartile range (IQR)]. Statistical significance was set at *p* < 0.05.

## 3. Results

In total, 137 patients were screened between October 2020 and June 2021. Of these, 18 patients did not meet the inclusion criteria and 5 patients declined to participate. The remaining 114 patients were randomly allocated to the NG group (57 patients) or the SC group (57 patients) for statistical analysis (Figure 2).

There was no difference between the two groups in terms of patient demographics, pre-operative airway evaluations, and intubation variables (Table 1).

The total intubation time was significantly shorter in the SC group than in the NG group (45.3 ± 10.0 s vs. 52.4 ± 9.7 s, *p* < 0.001). The insertion time from the nostril to the oropharynx was 18.0 ± 5.4 s in the SC group and 23.1 ± 5.3 s in the NG group and thus, significantly shorter in the SC group (*p* < 0.001, Table 2).

When the nasotracheal tube was passing through the nose, the number of preformed nasotracheal tube manipulations, navigability, and tube rotation did not differ significantly between the two groups. Regarding the passage of the nasotracheal tube through the nasal cavity into the pharynx, there were no differences between the two groups in the degree of nasal bleeding through direct laryngoscopy, the use of Magill forceps, or the total number of intubation attempts. Endotracheal intubation was successful in all patients, and there was no difference between the two groups in the direction of the tube in the nasal cavity (upper pathway or lower pathway) or the total duration of anesthesia. In the PACU, the nasal pain score and persistent epistaxis were similar between the two groups. A comparison of the outcomes between the two groups is shown in Table 2.

For the stress relaxation test, the force required to return the guiding devices to the original state was greater for the suction catheter (9.01592 MPa) than that for the nasogastric tube (4.07734 MPa) (Figure 3).

Comparing the force required to remove the aids, the removal of the NG tube required more force than the removal of the suction catheter (Table 3).

## 4. Discussion

When we compared the results of the two groups, the total intubation time was shorter in the SC group than in the NG group. However, there were no differences in the number of intranasal manipulations, navigability, or epistaxis. These results suggest that the suction-catheter-guided method is useful for nasotracheal intubation. Using a suction catheter as a guiding device can shorten the time required to complete nasotracheal intubation without additional side effects relative to the previously reported nasogastric tube-guided method.

During conventional nasotracheal intubation, complications such as epistaxis, nasotracheal tube obstruction, bacteremia, perforation of the pyriform fossa, sinusitis, etc., may occur as the intubation tube passes through the nasopharynx [2]. In particular, epistaxis is a common complication. The nasal mucosa throughout the nasal cavity is rich in blood vessels; thus, bleeding can occur if any part of the nasal cavity is injured [1,11]. When epistaxis occurs, the visual field is obscured during endotracheal intubation, and if massive nasal bleeding occurs, even mask ventilation may be difficult [12]. Moreover, traumatic damage to the structures of the nasal cavity and nasopharynx may also occur occasionally. Injuries to the nasal cavity and/or the nasopharynx lead to infection [13] and nasotracheal tube obstruction. In severe cases, ventilation may be compromised [14]. Therefore, it is essential to prevent and minimize these complications during nasotracheal intubation.

Numerous techniques have been tried to reduce the potential damage that can be caused by nasotracheal intubation [3,4,5,6,7,8,9,10]. These include the use of small-sized tubes, repositioning of the bevels, the use of thermo-softening tubes, and tube guidance methods. Of these, the tube guidance methods have shown a higher success intubation rate and fewer complications than the conventional blind method for nasotracheal intubation. The most common cause of epistaxis during nasotracheal intubation is most likely due to injuries to the surrounding soft tissue while passing the tube through the narrow nasal passage due to the hard end of the tube. Some papers have reported less epistaxis using thermos-softening tubes [4] and the Parker Flex-Tip™ tube (rounded with a flexible tapered distal tip) [15]. Previous studies that investigated aids for guiding the nasotracheal tube reported that epistaxis was reduced when using the aids compared to using only the nasotracheal tube [3,5,7,9,10]. In our study, the incidence of epistaxis was 35.1% in the NG tube group and 43.9% in the SC group, respectively. Our results are lower than the 60–80% rates reported in conventional blind groups in previous studies. [3,7,9]. As the nasogastric tube made of silicon is more flexible and softer than the suction catheter, which is made of PVC, it is thought that epistaxis is less likely to occur, but interestingly, the results showed that there was no difference in the incidence of epistaxis between the two groups. Perhaps the cause of nasal bleeding is the entry of the tube itself rather than the guiding device.

During nasotracheal intubation, the insertion time from the nostril to the oral cavity was shorter in the SC group than NG group. Moreover, the total intubation time was also shorter in the SC group than in the NG group. These results may be attributed to the physical characteristics of the materials of the two guiding aids. It is presumed that the suction catheter is thinner and more rigid than the nasogastric tube (Figure 3), so it easily passes through the nasal cavity and is easily removed from the preformed nasotracheal tube and with less resistance (Table 3). However, there is a bigger gap between a thinner aiding catheter and the intubating tube may increase the chance of impingement and nasal mucosa injury. So, it is considered that further studies involving a larger number will be needed.

When the nasotracheal tube passes through the nasal cavity, it passes either through the lower pathway between the floor of the nose and the inferior turbinate or the upper pathway between the inferior turbinate and the middle turbinate. It is known that complications occurring during nasotracheal intubation are more likely when passing via the upper pathway where major structures, such as the middle turbinate and the cribriform plate, are located. To avoid these structures, it is safe for the nasotracheal tube to enter via the lower pathway [2]. The preformed tube, which is often used for nasotracheal intubation, has a natural curvature that enables the tube to enter the nostril to the laryngeal inlet easily, but it has been reported that due to the curvature, these tubes mainly pass through the upper pathway [16]. Lim et al. [9] have shown that when nasotracheal intubation was performed using a nasogastric tube, the rate of passage of the tube to the lower pathway was higher than that of the group using the conventional technique. It was assumed that the flexible nature of the nasogastric tube facilitates its passage via the lower pathway. In this study, we checked with a fiberoptic endoscope to assess the nasal passage through which the preformed nasotracheal tube entered when using the two guiding aids. If there is a tube in the nasal cavity, it may be difficult to enter the fiberoptic endoscope due to the narrow space. In particular, if the gap between the nostril and the tube is narrow or if there is a deviated septum, the possibility of entry becomes difficult. In our experiment, entry of the fiberoptic endoscope was successful in all patients and it could be evaluated the nasal passage into which the preformed nasotracheal tube. The results indicated that there were no differences in the rate of passing into the lower pathway between the two groups (Table 2). Since the SC catheter has excellent restorative power (Figure 3), it was thought that it would be straighter when guiding the tube after passing through the nasal cavity. This result is presumably due to a force greater than the restoring force acting when the tube passes the guiding device.

This study has several limitations. First, nasotracheal intubation is a procedure that is highly dependent on the experience and skill of the practitioners. To reduce the variation of procedures among practitioners, one anesthesiologist performed all the intubations in this study. Hence, this may limit the generalizability of our results. To validate this technique, further studies are warranted. Second, we did not check whether the guidance aids were in the upper and lower pathways in the nasal cavity. We checked the position of the nasotracheal tube after intubation was completed. Thus we cannot exclude the possibility that the nasotracheal tube may have moved when the aid was removed from the nasotracheal tube. Lastly, since it is not possible to distinguish whether the epistaxis was caused by the aid or the nasotracheal tube, it is not known which aid is less invasive.

## 5. Conclusions

In conclusion, the use of a guiding aid for nasotracheal intubation is efficient. In particular, a suction catheter can be useful because it shortens the intubation time and does not increase complications.

## Figures and Tables

**Figure 1 ijerph-20-04503-f001:**
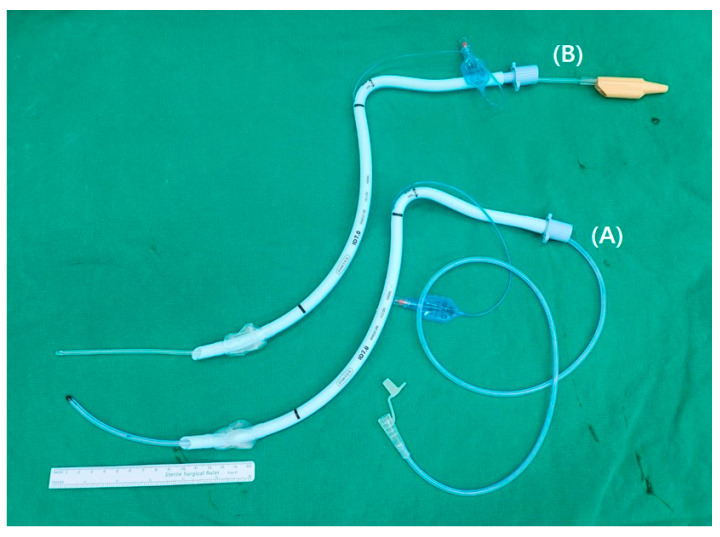
Preformed nasotracheal tubes with guiding devices: (**A**) Nasogastric tube. (**B**) Suction catheter.

**Figure 2 ijerph-20-04503-f002:**
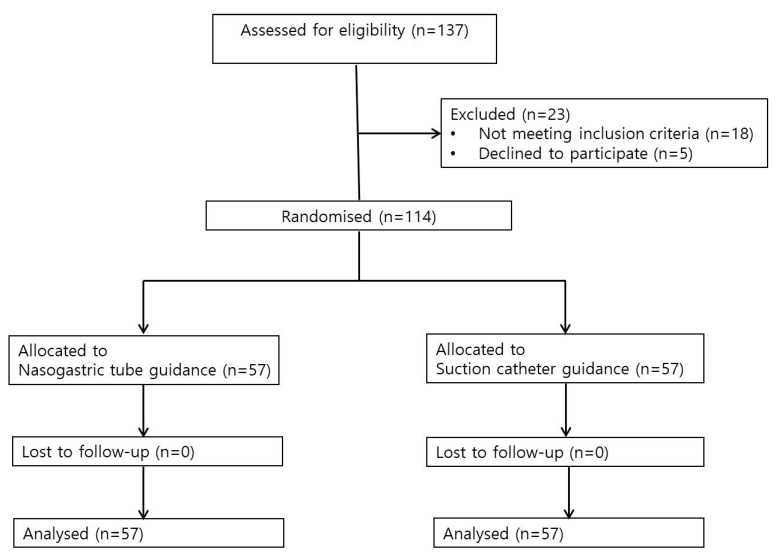
Study flow diagram.

**Figure 3 ijerph-20-04503-f003:**
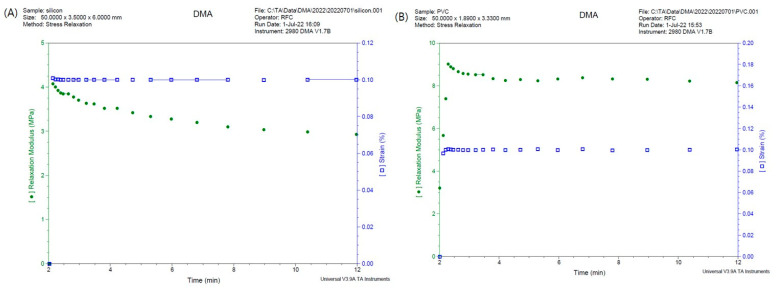
Stress relaxation test results using the CMA 2980 Dynamic Mechanical Analyzer: (**A**) Nasogastric tube. (**B**) Suction catheter.

**Table 1 ijerph-20-04503-t001:** Patient demographics, pre-operative airway examinations, and intubation variables.

Variable	NG Group (*n* = 57)	SC Group (*n* = 57)	*p* Value
Age, years	41.0 ± 16.8	42.2 ± 18.0	0.728
Gender			0.561
Female	19 (33.3%)	23 (40.4%)	
Male	38 (66.7%)	34 (59.6%)	
ASA physical status			0.077
I	5 (8.8%)	6 (10.5%)	
II	47 (85.2%)	44 (77.2%)	
III	5 (8.8%)	7 (12.3%)	
IV	0 (0%)	0 (0%)	
Height, cm	167.7 ± 7.2	165.6 ± 10.4	0.223
Body weight, kg	68.0 ± 11.3	64.6 ± 12.5	0.131
Neck circumference, cm	39.3 ± 3.6	38.3 ± 3.9	0.170
Thyromental distance, cm	7.9 ± 1.0	8.0 ± 1.1	0.477
Mallampati class			0.842
I	45 (78.9%)	46 (80.7%)	
II	10 (17.5%)	10 (17.5%)	
III	2 (3.5%)	1 (1.8%)	
IV	0 (0%)	0 ( 0%)	
Nasotracheal tube size (ID, mm)			0.267
7	29 (50.9%)	21 (36.8%)	
6.5	15 (26.3%)	22 (38.6%)	
6	13 (22.8%)	14 (24.6%)	
Intubated nostril site			1.000
Right	42 (73.7%)	42 (73.7%)	
Left	15 (26.3%)	15 (26.3%)	
Cormack grade			0.480
I	32 (56.1%)	34 (59.6%)	
II	20 (35.1%)	15 (26.3%)	
III	5 (8.8%)	8 (14.0%)	
IV	0 (0%)	0 (0%)	
“BURP” maneuver			0.809
Yes	10 (17.5%)	11 (19.3%)	
No	47 (82.5%)	46 (80.7%)	
Neck extension limitation			0.679
Yes	4 (7%)	2 (3.5%)	
No	53 (93.0%)	55 (96.5%)	

Values are presented as mean ± standard deviation or number of patients (percentage within the group or number of events). *p* < 0.05 is considered significant. Abbreviations: ASA, American Society of Anesthesiologists. ID, Internal Diameter. BURP, Backward, Upward, and Right-sided Pressure.

**Table 2 ijerph-20-04503-t002:** Outcome variables.

Variable	NG Group (*n* = 57)	SC Group (*n* = 57)	*p* Value
Number of manipulations in nasal cavity			0.580
1	45 (78.9%)	47 (82.5%)	
2	11 (19.3%)	10 (17.5%)	
3	1 (1.8%)	0 (0%)	
Navigability			0.461
Smooth	29 (50.9%)	24 (42.1%)	
Resistance	25 (43.9%)	27 (47.4%)	
Obstructive	3 (5.3%)	6 (10.5%)	
Tube rotation			0.206
Yes	5 (8.8%)	1 (1.8%)	
No	52 (91.2%)	56 (98.2%)	
Degree of nasal bleeding at intubation			0.618
No	37 (64.9%)	32 (56.1%)	
Mild	18 (31.6%)	22 (38.6%)	
Moderate	2 (3.5%)	3 (5.3%)	
Severe	0 (0%)	0 (0%)	
Use of Magill forceps			0.453
Yes	29 (50.9%)	24 (42.1%)	
No	28 (49.1%)	33 (57.9%)	
Trial of intubation			1.000
1st	53 (93.0%)	53 (93.0%)	
2nd	4 (7.0%)	4 (7.0%)	
Nostril to oropharynx time, s	23.1 ± 5.3	18.0 ± 5.4	0.000
Total intubation time, s	52.4 ± 9.7	45.3 ± 10.0	0.000
Tube pathway of nasal cavity			0.425
Upper pathway	26 (45.6%)	24 (42.1%)	
Lower pathway	31 (54.4%)	33 (57.9%)	
Duration of anesthesia, min	142.5 ± 65.8	131.1 ± 67.9	0.361
Complications in PACU			
Nasal pain score	2.2 ± 1.3	1.9 ± 1.3	0.321
Persistent Epistaxis			0.496
Yes	0 (0%)	2 (3.5%)	
No	57 (100%)	55 (96.5%)	

Values are presented as mean ± standard deviation or number of patients (percentage within the group or number of events). *p* < 0.05 is considered significant. Abbreviations: PACU, Post-Anesthesia Care Unit.

**Table 3 ijerph-20-04503-t003:** The force when removing the aids.

Nasotracheal Tube Size (ID, mm)	Nasogastric Tube 12Fr	Suction Catheter 10Fr
6.0	2.45 ± 0.286	1.06 ± 0.289
6.5	1.65 ± 0.213	0.804 ± 0.145
7.0	1.43 ± 0.396	0.549 ± 0.054

Values are presented as mean ± standard deviation (Unit: N = kg × 9.80665). Abbreviations: ID, Internal Diameter.

## Data Availability

The datasets generated and/or analyzed during the current study are not publicly available due to Hospital internal regulations but are available from the corresponding author upon reasonable request.

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
