# Peer review of "Comparison of the Nasal Cavity Guidance Methods’ Effects during Nasotracheal Intubation Using a Preformed Nasotracheal Tube: A Prospective Randomized Controlled Trial"

_ijerph, 2023, doi:10.3390/ijerph20054503_

Round 1

Reviewer 1 Report

The problem discussed in the manuscript is not very important in anesthesia management and has little impact on the overall result, however, the study was very clearly and interestingly presented.

1.     In Abstract, the abbreviations SC and NC should be used for the first time in line 23.

2.     In Table 2, mean and SD are missing from the legend.

3.     What a numerical rating scale was used to assess nasal pain?

4.     How long was the patient in PACU and what does persistent epistaxis mean according to the authors?

5.     In my version of the manuscript, the rows in the tables are not on the same line and this makes it difficult to read.

Author Response

  • We appreciate the reviewer’s sincere criticism!

  1. In Abstract, the abbreviations SC and NC should be used for the first time in line 23.
  • We have modified that the abbreviations SC and NC are used for the first time in line 23.

  1. In Table 2, mean and SD are missing from the legend.
  • We have added mean and SD descriptive text to the legend.

  1. What a numerical rating scale was used to assess nasal pain?
  • We assessed by the Numeric Rating Scale (NRS).
  • The Numeric Rating Scale (NRS) is the simplest and most commonly used numeric scale in which rates the pain from 0 (no pain) to 10 (worst pain).

  1. How long was the patient in PACU and what does persistent epistaxis mean according to the authors?
  • Patients usually stayed in the PACU for about an hour and were moved to ward if they were included the conditions for Post-Anesthetic Recovery Score.
  • We considered that persistent epistaxis is if nasal bleeding continued in the PACU after extubation of the nasotracheal tube.
  • We have modified and added.

  1. In my version of the manuscript, the rows in the tables are not on the same line and this makes it difficult to read.
  • We will modify and upload it again.

Reviewer 2 Report

 After carefully reading this paper some issues arise, in my opinion, and I believe the article should be considered for publication in its present form only after addressing these issues and either correcting or explaining them:

  • In the Introduction, lines 50-53: I cite "A nasogastric tube is made of silicone and is more flexible than a suction catheter, but the diameter of the nasogastric tube is thicker. A suction catheter is made of poly- vinyl chloride (PVC) and is relatively thin and less flexible compared with nasogastric tube"- not all nasogastric tubes are made of silicone, and not all suction catheters are made of PVC. This statement partly alters your hypothesis, since many nasogastric tubes are made of PVC, similar to suction catheters. Please correct that phrase and state that these are the types of catheters you tested in your study as other institutions may have different material suction or nasogastric catheters. Also, the nasogastric tubes and suction catheters come in a wide variety of sizes so it is not correct to state that one is thicker than the other. Please change this paragraph!

  • In the Methods:

1. how exactly did you randomize the patients? 

2. You should specify the length of the suction catheter, nasogastric tube, and intubating tubes as this may have some importance in the procedure. Also, specify if the intubating tube diameter is measured internally or externally.

3. Please explain why don't you use any nasal decongestant (vasoconstrictor) in preparation for nasal intubation. Line 118 - Please correct Margill with Magill.

4. Please describe the procedure in some more detail: for example how was the intubating tube bevel oriented during the maneuver and the rotation direction in case an obstacle was encountered? Was it planned in case of failure to change the nostril? Who helped the anesthesiologist to perform the procedure?

5. Line 125/ 126: I cite "After tracheal intubation was completed, the tube location in the upper or lower pathway of the nasal cavity was checked using a fiberoptic endoscope". How was this achieved in a patient intubated through the nose? Was there enough space in the nostril to accommodate both the fiberscope and the intubating tube? Please explain. 

  • In the Discussion:
  1. I cite: "Using a suction catheter as a guiding device can shorten the time required to complete nasotracheal intubation without additional side effects relative to the previously reported nasogastric tube guided method [9]" - please specifically name those side effects. I read the article you reference for this statement and it is not clear which are those side effects. 
  2.  I can not agree that there is no difference regarding the epistaxis occurrence in those two groups- lines 236/237. There is a 20-25% difference favoring the nasogastric tube which did not return a statistical significance. Also please cite the study with a 60-80% epistaxis incidence- lines 233/234 . 
  3. Although the insertion time and the extraction from the intubating tube are better with a thinner catheter (lines 242-244), I think you should mention that a bigger gap between a thinner aiding catheter and the intubating tube may increase the chance of impingement and nasal injury.
  4. Please present here the possible limitations of the fiberoptic maneuver that revealed the intubating tube position inside the nasal cavity. And if some patients were difficult or impossible to examine. 

Author Response

  • We appreciate the reviewer’s sincere criticism!

  • In the Introduction, lines 50-53: I cite "A nasogastric tube is made of silicone and is more flexible than a suction catheter, but the diameter of the nasogastric tube is thicker. A suction catheter is made of poly- vinyl chloride (PVC) and is relatively thin and less flexible compared with nasogastric tube"- not all nasogastric tubes are made of silicone, and not all suction catheters are made of PVC. This statement partly alters your hypothesis, since many nasogastric tubes are made of PVC, similar to suction catheters. Please correct that phrase and state that these are the types of catheters you tested in your study as other institutions may have different material suction or nasogastric catheters. Also, the nasogastric tubes and suction catheters come in a wide variety of sizes so it is not correct to state that one is thicker than the other. Please change this paragraph!
  • ⇒Thanks to the reviewer for the comments. We admit that described the sentences without thinking deeply about the material of SC or NG. So, we have modified as follows.⇒Several methods have been proposed to facilitate tube passage and minimize the complications of nasotracheal intubation [2-10]. Prior to intubation, it is necessary to carefully evaluate the anatomical variations and conditions of the nasal and oral cavities, and select a suitable sized tube for the patient. The use of an appropriate guiding device may further facilitate intubation [2]. A guiding device that easily passes the nasal cavity has the advantage of enabling the smooth passage of the nasotracheal tube, while reducing the potential intranasal damage. Each of these guiding devices has different physical and morphological characteristics. Among them, nasogastric tubes and suction catheters are easily available in operating rooms. Both have a long thin shape and good flexibility, which allows them to easily pass from the nasal cavity into the oropharynx.  We hypothesized that differences in intubation conditions may occur due to the different physical and morphological characteristics of these guiding devices. Therefore, this study aimed to investigate and compare the intubation conditions facilitated by these two guiding devices for nasotracheal intubation.
  • In the Methods:
  1. how exactly did you randomize the patients?

⇒The enrolled patients were randomly allocated to either the NG group or SC group, using a computer-generated randomization code, the allocation rate was 1:1.

⇒We have added this part.

  1. You should specify the length of the suction catheter, nasogastric tube, and intubating tubes as this may have some importance in the procedure. Also, specify if the intubating tube diameter is measured internally or externally.

⇒As pointed out by the reviewer, we also thought that the length of the guiding device was important in this study. If the guiding device is too long, it is difficult to remove it, and if it is too short, there is a possibility that the tube cannot be guided. As shown in Figure 1., the preformed nasotracheal tube was threaded over the prepared nasogastric tube (Figure 1-A) or the suction catheter (Figure 1-B) and protruded by 10 cm from the distal end of the preformed nasotracheal tube.

⇒We selected the intubation tube diameter based on the internal diameter (ID). Commonly, female selected an ID of 6.5 mm and male, ID of 7.0 mm, and when the patient's nostril diameter was smaller, a 0.5 mm smaller tube was selected.

⇒We have modified and added.

  1. Please explain why don't you use any nasal decongestant (vasoconstrictor) in preparation for nasal intubation. Line 118 - Please correct Margill with Magill.

⇒Epinephrine, phenylephrine or oxymetasoline are mainly used as vasoconstrictors before nasotracheal intubation [1] , but we have experienced hemodynamic unstability after using epinephrine before. There have been a case report of similar occurrences of tachycardia [2], so our hospital policy does not use vasoconstrictor as pretreatment.

  1. Park, D.H.; Lee, C.A.; Jeong, C.Y.; Yang, H.S. Nasotracheal intubation for airway management during anesthesia. Anesthesia and pain medicine 2021, 16, 232-247, doi:10.17085/apm.21040.
  2. Hoshi, T.; Suzuki, T.; Somei, M.; Iijima, T.; Kurihara, Y. Sudden Tachycardia Due to Submucosal Migration of an Epinephrine-Soaked Swab During Nasal Intubation. Anesthesia progress 2018, 65, 259-260, doi:10.2344/anpr-66-01-02.

⇒We modified with Magill.

  1. Please describe the procedure in some more detail: for example how was the intubating tube bevel oriented during the maneuver and the rotation direction in case an obstacle was encountered? Was it planned in case of failure to change the nostril? Who helped the anesthesiologist to perform the procedure?

⇒Inserted a nasogastric tube or a suction catheter together with a preformed nasotracheal tube into the nostril, and entered it along the nasal floor toward the posterior nasopharyngeal wall

⇒Endotracheal intubation was performed as follows. First, A guidance device advanced 10 cm beyond the tube is introduced into the selected nostril. After entering the guidance device, adjust the bevel of the tube to be in the opposite direction to the nasal septum, and then enter the nasal cavity. At this time, if the curve of the tube is directed toward the head, proceed 5 to 7 cm, rotate 180 degrees, and then proceed further.

⇒During intubation, a preformed nasotracheal tube was first inserted into the lower pathway (1st manipulation). If it was not possible to enter due to resistance, the tube was redirected toward more cauded (2nd manipulation). If it still did not enter, redirected the tube facing slightly cephalad (3rd manipulation).

⇒The content pointed out by the reviewer is described in the previously submitted document, but if there is additional content to be described, please mention it in detail and we will make further corrections.

  1. Line 125/ 126: I cite "After tracheal intubation was completed, the tube location in the upper or lower pathway of the nasal cavity was checked using a fiberoptic endoscope". How was this achieved in a patient intubated through the nose? Was there enough space in the nostril to accommodate both the fiberscope and the intubating tube? Please explain.

⇒We performed it as shown in the following figures.

⇒After nasotracheal intubation, it was confirmed that there was a space to insert a fibroscope in the space next to it.

  • In the Discussion:
  1. I cite: "Using a suction catheter as a guiding device can shorten the time required to complete nasotracheal intubation without additional side effects relative to the previously reported nasogastric tube guided method [9]" - please specifically name those side effects. I read the article you reference for this statement and it is not clear which are those side effects.

⇒Really sorry. It was a phrase of the results of our experiment, but the reference was entered incorrectly.

⇒We have modified.

  1. I can not agree that there is no difference regarding the epistaxis occurrence in those two groups- lines 236/237. There is a 20-25% difference favoring the nasogastric tube which did not return a statistical significance.

⇒In the previous reports, the incidence of epistaxis was shown lower in the guiance groups compared to the conventional blind group. In our experiment, the two guidance groups were compared, and there seems to be no difference in the incidence of epistaxis between the two groups.

Also please cite the study with a 60-8⇒0% epistaxis incidence- lines 233/234 .

⇒In each reports, 86%, 53.6%, and 86.7% of the conventional blind group were reported.

⇒Our results were shown lower than the 60% - 80% incidence of epistaxis of the conventional blind method reported in previous studies.

⇒We have modified and added references.

  1. Although the insertion time and the extraction from the intubating tube are better with a thinner catheter (lines 242-244), I think you should mention that a bigger gap between a thinner aiding catheter and the intubating tube may increase the chance of impingement and nasal injury.

⇒We totally agree with the reviewer's opinion.

⇒However, there is a bigger gap between a thinner aiding catheter and the intubating tube may increase the chance of impingement and nasal injury. So, it is considered that further studies involving a larger number will be needed.

⇒We have modified and added.

  1. Please present here the possible limitations of the fiberoptic maneuver that revealed the intubating tube position inside the nasal cavity. And if some patients were difficult or impossible to examine.

⇒There are possible limitations of fiber optic manipulation for positioning the intubation tube inside the nasal cavity. If there is a tube in the nasal cavity, it may be difficult to enter the fiber due to the narrow space. In particular, if the gap between the nostril and the tube is narrow or if there is a deviated septum, the possibility of entry becomes difficult.

Reviewer 3 Report

What was the size of ETT and catheter/nasogastric tube ?

Whether selection of catheter or nasogastric tube size was different or same for different ETTs. If different then what was the criteria to choose. If same then compare the bias. Clarification is required to interpret the bias.

Author Response

  • We appreciate the reviewer’s sincere criticism!

What was the size of ETT and catheter/nasogastric tube?

Whether selection of catheter or nasogastric tube size was different or same for different ETTs. If different then what was the criteria to choose. If same then compare the bias. Clarification is required to interpret the bias.

  • We selected the intubation tube diameter based on the internal diameter (ID). Commonly, female selected an ID of 6.5 mm and male, ID of 7.0 mm, and when the patient's nostril diameter was smaller, a 0.5 mm smaller tube was selected.
  • There was no difference in the size of the nasotracheal tube between the two groups (Table 1).
  • The nasogastric tube of 12 Fr and the suction catheter of 10 Fr were used equally.

Round 2

Reviewer 2 Report

I am largely satisfied with the changes operated in the article,  except for the possible limitations of the fiberoptic maneuver to reveal the intubating tube position inside the nasal cavity, which did not find a place in the Discussion section. The manuscript looks suitable for publication to me if this matter is also covered.

Author Response

  • We appreciate the reviewer’s sincere criticism again!

  • I am largely satisfied with the changes operated in the article, except for the possible limitations of the fiberoptic maneuver to reveal the intubating tube position inside the nasal cavity, which did not find a place in the Discussion section. The manuscript looks suitable for publication to me if this matter is also covered.
  • We have added the paragraph to the discussion.
  • When the nasotracheal tube passes through the nasal cavity, it passes either through the lower pathway between the floor of the nose and the inferior turbinate or the upper pathway between the inferior turbinate and the middle turbinate. It is known that complications occurring during nasotracheal intubation are more likely when passing via the upper pathway where major structures, such as the middle turbinate and the cribriform plate, are located. To avoid these structures, it is safe for the nasotracheal tube to enter via the lower pathway [2]. The preformed tube, which is often used for nasotracheal intubation, has a natural curvature that enables the tube to enter the nostril to the laryngeal inlet easily, but it has been reported that due to the curvature, these tubes mainly pass through the upper pathway [16]. Lim et al [9]. have shown that when nasotracheal intubation was performed using a nasogastric tube, the rate of passage of the tube to the lower pathway was higher than that of the group using the conventional technique. It was assumed that the flexible nature of the nasogastric tube facilitates its passage via the lower pathway. In this study, we checked with fiberoptic endoscope to assess the nasal passage through which the preformed nasotracheal tube entered when using to the two guiding aids. If there is a tube in the nasal cavity, it may be difficult to enter the fiberoptic endoscope due to the narrow space. In particular, if the gap between the nostril and the tube is narrow or if there is a deviated septum, the possibility of entry becomes difficult. In our experiment, entry of the fiberoptic endoscope was successful in all patients and it could be evaluated the nasal passage into which the preformed nasotracheal tube. The results indicated that there were no differences in the rate of passing into the lower pathway between the two groups (Table 2).